# Plant Phenolics and Extracts in Animal Models of Preeclampsia and Clinical Trials—Review of Perspectives for Novel Therapies

**DOI:** 10.3390/ph14030269

**Published:** 2021-03-16

**Authors:** Marcin Ożarowski, Tomasz M. Karpiński, Michał Szulc, Karolina Wielgus, Radosław Kujawski, Hubert Wolski, Agnieszka Seremak-Mrozikiewicz

**Affiliations:** 1Department of Biotechnology, Institute of Natural Fibres and Medicinal Plants, Wojska Polskiego 71b, 60-630 Poznań, Poland; karolina.wielgus@iwnirz.pl; 2Chair and Department of Medical Microbiology, Poznań University of Medical Sciences, Wieniawskiego 3, 61-712 Poznań, Poland; tkarpin@ump.edu.pl; 3Department of Pharmacology, Poznań University of Medical Sciences, Rokietnicka 5a, 60-806 Poznań, Poland; mszulc@ump.edu.pl (M.S.); radkuj@ump.edu.pl (R.K.); 4Division of Gynecology and Obstetrics, Podhale Multidisciplinary Hospital, 34-400 Nowy Targ, Poland; hubertwolski@wp.pl; 5Division of Perinatology and Women’s Diseases, Poznań University of Medical Sciences, Polna 33, 60-535 Poznań, Poland; asm@data.pl; 6Laboratory of Molecular Biology in Division of Perinatology and Women’s Diseases, Poznań University of Medical Sciences, 60-535 Poznań, Poland; 7Department of Pharmacology and Phytochemistry, Institute of Natural Fibres and Medicinal Plants, 62-064 Poznań, Poland

**Keywords:** phenolic compounds, plant extracts, in vivo models of preeclampsia, clinical trials

## Abstract

The current health requirements set the direction in pharmacological research, especially as regards diseases that require improvement of existing therapeutic regimens. Such diseases include preeclampsia, which is a hypertensive disorder of pregnancy during which there occurs progressive increasing activation of the immune system through elevation of pro-inflammatory cytokines and antiangiogenic factors, which is dangerous for the mother and fetus. A promising field of research for new drugs to treat this disease is the study of natural phenolic compounds of plant origin and herbal extracts, which are complex matrices of chemical compounds with broad biological activities. Many plant substances with anti‑inflammatory and anti‑hypertensive properties are known, but studies in animal models of preeclampsia and clinical trials concerning this disease constitute a new and developing research trend of significant medical importance. The aim of our research review was to identify and analyze the results of already available studies on baicalin, curcumin, epigallocatechin gallate, punicalagin, quercetin, resveratrol, salvianolic acid A (danshensu), silibinin, and vitexin, as well as plant extracts from *Brassica oleracea* L., *Euterpe oleracea* Mart., *Moringa oleifera* Lam., *Punica granatum* L., *Silybum marianum* (L.) Gaertner, *Thymus schimperi* Ronniger, *Uncaria rhynchophylla* (Miq.) Miq. ex Havil., and *Vitis vinifera* L., which are potential and promising candidates for further research and for potential new therapies.

## 1. Introduction

Preeclampsia (ICD-10 code 014) is a multifactorial pregnancy disorder, which is globally dangerous to the health and life of the mother and the fetus [1,2]. In addition, there is currently a lack of choice in treatments for this complicated disease. A potential route to the development of a new drug or drug combination is to elucidate the pathogenesis of preeclampsia. This has been discussed in many research papers that emphasize the fundamental role of the inflammatory process [2,3,4,5,6]. On the other hand, a promising source of available materials for chemical modifications in searching for new medicinal drugs may be plant phenolic compounds and extracts as multi-component mixtures, which show anti-oxidant, anti-inflammatory, and anti-hypertensive properties. Assessment of the effects of plant phenolic compounds in animal models relevant to preeclampsia have been insufficiently reported, but the first studies which raise hope for new optional medicines for the treatment of preeclampsia have been published. Our review of the largest scientific databases (PubMed, Medline, Scopus) over the past 10 years (or earlier) showed that only nine natural chemical compounds from the group of plant phenolics, in 15 studies, used animal models of preeclampsia and three clinical trials and three ex vivo studies were carried out. The compounds most intensively investigated in this field are resveratrol, quercetin, curcumin, salvianolic acid A (danshensu), baicalin, epigallocatechin gallate, punicalagin, silibinin, and vitexin. Furthermore, plant extracts, as mixtures of various chemical compounds, were also investigated in preeclampsia. Eight promising plant extracts—*Euterpe oleracea* Mart., *Moringa oleifera* Lam., *Punica granatum* L., *Thymus schimperi* Ronniger, *Uncaria rhynchophylla* (Miq.) Miq. ex Havil., *Vitis vinifera* L.—were tested in animal models of preeclampsia (in all seven studies), and *Brassica oleracea* L. and *Silybum marianum* (L.) Gaertner were investigated in two clinical trials concerning preeclampsia (one clinical trial in phase III is in progress). Together, we carried out an analysis of 30 studies published only in English.

These studies are very much needed because drug treatment options in mild and severe preeclampsia are limited. Only methyldopa is used as the first-line treatment of this disease [2]. The analysis of their results can build future solutions and open up new pharmacological possibilities.

## 2. General Characteristics of Plant Phenolics Tested in Preeclampsia

Plant phenolics are a large group of natural chemical compounds with various chemical structures, which are widely distributed in plants in all parts of the world [7,8]. According to Dai et al. [7] structures of more than 8000 phenolic compounds have been described, and biological and pharmacological properties are constantly examined. However, it should be noted that only some of them have been widely studied in clinical trials. Phenolic compounds with anti-oxidant, anti-inflammatory, and potential anti-hypertensive effects in relation to preeclampsia, include the following: baicalin, curcumin, epigallocatechin gallate, punicalagin, quercetin, resveratrol, salvianolic acid A, silibinin, and vitexin.

### 2.1. Baicalin

Baicalin (Figure 1) is the main flavonoid produced in underground parts of *Scutellaria baicalensis* Georgi [9], which exerts anti-inflammatory, cardioprotective, antiplatelet, anticoagulant, profibrinolytic, and anti-hypertensive effects [10,11,12]. To date, only one study has been carried out in an animal model of preeclampsia [13]. In vitro and ex vivo studies related to the activity of baicalin have underlined the anti-inflammatory activity, myorelaxant property, inhibition of renin activity, attenuation of chronic hypoxia-induced pulmonary hypertension, and promotion of embryo adhesion and implantation after use of baicalin [14].

### 2.2. Curcumin

Curcumin (Figure 1) is a well-known and intensively studied lipophilic polyphenol occurring in the rhizomes of turmeric (*Curcuma longa* L., *Zingiberaceae*), which shows many biological activities as a culinary spice and component of Asian diets. Moreover, it displays a wide range of pharmacological properties as a component of traditional herbal products [15,16]. Recently, consumption of curcumin is increasing not only in the prevention and treatment of various disturbances and diseases [15] but also in pregnancy [17]. Because information about the influence of curcumin on pregnancy and pregnancy-related complications is still limited, few experimental studies on preeclampsia models have been carried out [18,19]. Recently, Filardi et al. [17] and Ghaneifar et al. [20] summarized the current state of knowledge on the impact of curcumin on pathophysiological processes during pregnancy. Curcumin through its wide range of activities may have positive effects on the management of pregnancy-related disorders including preeclampsia but also for fetal growth disorders. Recently, two studies were carried out in animal models of preeclampsia [18,19].

### 2.3. Epigallocatechin Gallate

Epigallocatechin gallate (EGCG) (Figure 1) is a catechin type compound of *Camelia sinensis* (L.) Kuntze (*Theaceae*) [21]. Many studies have revealed that EGCG exerts strong anti-oxidant activity among other catechins [21,22]. Moreover, EGCG is well known to induce the production of nitric oxide from endothelial cells, and it may increase cardiovascular homeostasis [23]. The hypotensive effect of EGCG has been observed in experimental studies, which revealed that EGCG inhibited the activity of the angiotensin I-converting enzyme (ACE) [24,25,26].

### 2.4. Punicalagin

Punicalagin (Figure 1) is an ellagitannin, which is present in fruits of *Punica granatum* and is the most abundant polyphenolic in pomegranate juice [13,27]. It shows several biological and pharmacological activities due to its anti-oxidant, antimicrobial, anti-inflammatory, and anti-hyperglycemic properties [27]. Punicalagin exerted anti-hypertensive activities in a clinical study [13,28]. Previously, it was shown that this polyphenolic compound can attenuate apoptosis in the human placenta and in human placental trophoblasts [29,30]. Recently, the beneficial effect of punicalagin was observed in an animal model of preeclampsia by Wang et al. [13]. Additionally, besides punicalagin, all parts of *P. granatum* L. (*Lythraceae*) contain high levels of phenolic compounds, tannins, and anthocyanins [31,32]. Notably, various preparations of *P. granatum* exhibit many therapeutic properties towards cardiovascular diseases due to their anti-oxidant and anti-inflammatory activities, and they are considered as potential pro-healthy nutraceuticals [32]. Previously, it was shown that long-term use of the juice of *P. granatum* reduced blood pressure and lipid peroxidation of the LDL fraction in patients with carotid artery stenosis [33]. Cardioprotective effects of the juice were revealed in a clinical trial on patients with ischemic heart disease [34] and in hemodialysis patients [35]. Recently, many studies have proved the vasculoprotective effects of *P. granatum* in various models [13,36,37]. Interestingly, the results of in vitro studies demonstrated potential beneficial effects of the extract from fruits of *P. granatum* using human umbilical vein endothelial cells [38,39,40] and human placenta, and in human placental trophoblasts [29].

### 2.5. Quercetin

Quercetin (Figure 1) is a flavonol widely distributed, not only in green vegetables and fruits, but also in medicinal plants belonging to *Amaryllidaceae*, *Apiaceae*, *Hypericaceae*, *Lamiaceae*, *Rosaceae*, and *Passifloraceae*, among other families [41,42]. It is also commonly found (as such or as glycosidic derivatives) in plant-based food and beverages such as tea, fruit juices, wine, and honey [43,44]. According to the review of Ożarowski et al. [2], quercetin shows anti-hypertensive effects and protective effects against inflammation both in vitro, in vivo, and in clinical trials. Moreover, this flavonol exerts positive effects on the development of the embryo, fetus, and placenta without teratogenic and abortive effects [2]. However, recently only a few studies have been carried out in a preeclampsia model [45,46,47]. Anti-oxidant and anti-inflammatory effects of quercetin were previously summarized [2,48].

### 2.6. Resveratrol

Resveratrol (Figure 1) is a stilbene-based natural polyphenolic compound produced by various plants [49] belonging to various plant families, for example, *Ericaceae*, *Leguminosae*, *Liliaceae*, *Moraceae*, *Myrtaceae*, *Pinaceae*, *Polygonaceae*, *Rosaceae*, and *Vitaceae* [50,51]. The highest levels of resveratrol were detected in fruits of *Vitis vinifera* L. [52], *Passiflora edulis* L. [53], *Vaccinium macrocarpon* Aiton [54], *Vaccinium myrtillus* L., *Vaccinium vitis-ideae* L. var. *vitis-ideae*, *Vaccinium vitis-ideae* L. var. *minor*, and *Vaccinium angustifolium* Aiton [55]. In many studies, pharmacological activities are evaluated for their healthy properties not only in multi-targeting cardiovascular diseases [56] but also in cancers, degenerative and metabolic disorders [51,57,58,59]. Multi-directional biological and pharmacological properties of resveratrol are considered in the treatment of preeclampsia according to a few experimental studies and a clinical trial [60,61,62,63]. Recently, several pre-clinical studies have shown the anti-hypertensive effects of this polyphenolic compound focusing on mechanisms of action, that is, anti-oxidant potential, anti-inflammatory activity (by reducing the expression of NF‑κB, IL-6, and IL-1β), enhancing the production of nitric oxide in endothelium [61,64,65], improving markers of endothelial cells [66], and inhibiting vascular endothelial growth factor receptor-1 [61,67]. However, it should be noted that today the function of resveratrol is still poorly understood.

### 2.7. Salvianolic Acid A

Salvianolic acid A or danshensu (Figure 1), an active polyphenol, occurs in the roots of *Salvia miltiorrhiza* Bunge (*Lamiaceae*), which is one of the oldest and most valuable herbs used in Chinese folk medicine. This polyphenolic compound is used mainly in cardiovascular diseases because it shows antiplatelet, anticoagulant, and antithrombotic activities; reduces the effects of ischemia; and exerts anti-inflammatory effects [68,69,70]. Furthermore, not only extracts of *S. miltiorrhiza* but also salvianolic acid A can lower blood pressure [71]. Salvianolic acid was investigated in an animal model of preeclampsia by Shen et al. [72,73].

### 2.8. Silibinin

Silibinin (or silybin) (Figure 1) is a flavonolignan occurring in fruits and seeds of *Silybum marianum* (L.) Gaertner [74,75]. It is the main component of silymarin, well-documented as a hepatoprotective natural compound. Silibinin exhibits a wide spectrum of pharmacological activity in vitro, in vivo, and in clinical trials, mainly anti-oxidant and anti-inflammatory [75,76], anti-cancer and chemopreventive properties [77]. Recently, it was shown that silibinin can reduce systolic pressure, downregulating the expression of C-X-C chemokine receptor type 4 (CXCR4) in pulmonary arteries [78], upregulating the expression of CXCR4 in bone marrow [79], and inhibiting the production of cytokines (NF-κB, TNF-α, IL-1β) [80]. Lim et al. [76] observed that this flavonolignan can reduce the response of inflammatory pathways during infection-induced preterm birth. The activity of silibinin in experimental preeclampsia was studied by Souza et al. [81].

### 2.9. Vitexin

Vitexin (Figure 1) belongs to the group of C-glycosylated flavonoids and is produced in many plants from various botanical families, mainly in *Asteraceae* (*Matricaria* spp.) [82], *Lamiaceae* (*Rosmarinus* officinalis) [83], *Passifloraceae* (*Passiflora* spp.) [84], and *Rosaceae* (*Crataegus* spp., *Rosa* spp.) [85]. Vitexin shows a wide range of pharmacological activities, such as anti-hypertensive and anti-inflammatory effects and protective properties in the cardiovascular system [48,85]. Interesting results were obtained by Zheng et al. [86] in an animal model of preeclampsia.

## 3. Plant Phenolic Compounds in Animal Models of Preeclampsia

Recently, only Wang et al. [13] carried out a study on baicalin (50; 100 and 150 mg/kg/day, i.p. for 20 days) in experimental preeclampsia with dysfunction of kidneys and liver induced by the inhibitor of nitric oxide synthase—NG-nitro-L-arginine-methyl-ester (L-NAME). Baicalin dose-dependently reduced values of blood pressure and apoptosis of kidney and liver cells. Moreover, baicalin elevated the expressions of anti-apoptotic proteins such as XIAP and Bcl-2, and reduced expression of the apoptotic protein caspase-9 in the liver.

Gong et al. [18] examined the beneficial effect of curcumin (0.36 mg/kg), which was administered to pregnant rats with lipopolysaccharide (LPS)-induced preeclampsia. This study showed that this polyphenol decreased blood pressure and concentration of urinary protein. Curcumin can improve disorders caused by LPS such as deficient trophoblast invasion and remodeling of the spiral artery. Moreover, it was observed that curcumin diminished expression of TLR4 and NF-κB proteins in the placenta and IL-6 and MCP-1 proteins in rat serum, which were elevated by LPS. Because the toll-like receptor 4 signaling pathway takes part in the pathogenesis of inflammation during preeclampsia, the effect of curcumin observed in this study can explain its mechanism of action as an anti-inflammatory substance. In another study, Zhou et al. [19] observed in lipopolysaccharide-induced preeclampsia in pregnant mice that curcumin at a dose of 100 µg/kg/d i.g. also decreased expression of inflammatory factors such as TNF-α, IL-1β, IL-6 and levels of chemokines such as MCP-1 and MIP-1 and increased the level of phosphorylated Akt by its upregulation in the placenta. Moreover, curcumin was effective in inhibiting the infiltration of macrophages in this organ.

Punicalagin was investigated in pregnant rats with L-NAME induced hypertension and it was administered orally to animals at doses of 25, 50, or 100 mg/kg on days 14–21 of pregnancy. A comparative study revealed that the highest dose of punicalagin showed a stronger effect in decreasing systolic and diastolic blood pressure and also mean arterial pressure. A similar dose‑dependent effect was observed in enhancing the anti-oxidant capacity and increasing the levels of nitric oxide in the placenta. A molecular analysis led to the conclusion that punicalagin caused increased expression and levels of mRNA VEGF and downregulation of expression of vascular endothelial growth factor receptor-1/fms-like tyrosine kinase-1, suggesting that punicalagin has a positive effect on angiogenic balance during experimental preeclampsia [13].

Li et al. [45] observed that quercetin (2 mg/kg b.w.) administered to pregnant rats with LPS-induced preeclampsia significantly reduced systolic blood pressure, rescued abnormal uteroplacental angiogenic status, decreased the elevated changes of tyrosine kinase-1 (sFlt-1)/placental growth factor (PlGF) ratio, suppressed the production of cytokine production in the placenta (TNF α, IL-6 and MCP-1), reduced lipid peroxidation by reduction of the MDA level, and ameliorated adverse pregnancy outcomes. Because quercetin can significantly improve the pathophysiology of preeclampsia, it was concluded that this flavonoid may be a candidate for preeclampsia treatment. In another study, Yang et al. [46] investigated the effects of the interaction between acetylsalicylic acid (1.5 mg/kg b.w.) and quercetin (2 mg/kg b.w.), which were administered to rats with NG-nitro-L-arginine-methyl-ester-induced preeclampsia. These chemical substances were given from gestational day 4 to 19. The results showed that quercetin together with acetylsalicylic acid can enhance the beneficial effects of acetylsalicylic acid in rats mainly by decreasing the peroxidation of lipid and the level of cytokines in the plasma and placenta. Moreover, quercetin with acetylsalicylic acid reduced systolic blood pressure and expression levels of mRNA VEGF and mRNA sFlt-1. However, it was observed that quercetin given to rats alone had no effects on systolic blood pressure. A similar result was also reported previously when Tanir et al. [47] observed that quercetin (10 mg/kg b.w.) administered to rats with preeclampsia induced by the same substance (L-NAME) did not reduce the high blood pressure.

Resveratrol at a dose of 20 mg/kg/day i.g. was tested in a mouse model of preeclampsia induced by injection of L-NAME [60]. It was shown that resveratrol administered during the entire pregnancy did not significantly influence the number and birth weight of fetuses, the weight of the placenta, or external malformations in comparison with the control group. However, it was observed that this polyphenolic compound caused a significant reduction of blood pressure and level of protein in the urine, indicating the ameliorative effects of resveratrol on clinical phenotypes of preeclampsia. Moreover, Zou et al. [60] observed beneficial effects of resveratrol for the process of epithelial-mesenchymal transition in trophoblasts during the pathogenesis of preeclampsia through lowering the expression of E-cadherin and increasing the level of β-cadherin, N-cadherin, vimentin, snail, and MMP-2/MMP-9, and angiogenesis-related factors such as VEGF, sFlt-1, AngI, and AngII compared to the preeclampsia group not receiving resveratrol. Thus, Zou et al. [60] concluded that resveratrol may allow the stimulation of the invasive capability of human trophoblasts by promoting epithelial-mesenchymal transition and influencing the Wnt/β-catenin pathway during preeclampsia. In a previous study, Zou et al. [62] drew attention to the anti-oxidant properties of resveratrol (20 mg/kg per day, i.g. during the entire pregnancy), which was administered to rats with preeclampsia induced by L-NAME. Moreover, Zou et al. [62] reported anti-apoptotic effects in trophoblasts of the placenta of rats with experimental preeclampsia. Several studies have shown that oxidative stress may be involved in the pathogenesis of preeclampsia [87], and hence resveratrol should be considered in the prevention of oxidative stress. However, Moraloglu et al. [63] observed that resveratrol at doses of 40 mg/kg per day, i.g. administered during the entire pregnancy to rats with hypertension induced by injection of desoxycorticosterone acetate, did not decrease blood pressure or parameters of placental pathology.

Salvianolic acid A (10 μg/g and 30 μg/g, i.p.) administered to mice with preeclampsia induced by phosphatidylserine/phosphatidylcholine showed efficacy in ameliorating the prognosis of maternal and fetal syndrome. The results showed that polyphenolic acid at high and low doses exerted significant effects in reduction of thrombin time, similar to heparin, and increased the plasma antithrombin III activity equally as did acetylsalicylic acid. However, at high doses salvianolic acid A can decrease blood pressure and proteinuria. On the other hand, long-term use of low-dose salvianolic acid exerted better efficacy, possibly by anticoagulant activity. It was observed that salvianolic acid A in a high dose was more effective in decreasing blood pressure proteinuria to the normal level [72]. Moreover, Shen et al. [72] reported that salvianolic acid A at two doses of injection (10 μg/g and 30 μg/g) used in a similar model of preeclampsia in mice induced by phosphatidylserine/phosphatidylcholine (PS/PC) microvesicle administration, showed dose‑dependent effects. Salvianolic acid A at the highest dose demonstrated effective reduction of blood pressure and proteinuria to normal levels, and it elevated the platelet counts and lowered the expression of thrombomodulin in the placenta. Furthermore, this natural chemical compound at two different doses increased the activity of the plasma antithrombin III. However, low-dose salvianolic acid A decreased the level of D-D dimer. Considering all the evidence, Shen et al. suggest the long-term application of low-dose salvianolic acid A in pregnancy, which can bring an improvement of preeclampsia syndrome.

In a similar model of human preeclampsia induced by L-NAME, silibinin was administered orally to rats at a dose of 100 mg/kg/day by 10 days. In this study it was observed that silibinin can prevent the elevation of systolic blood pressure, can reduce the levels of pro-inflammatory factors TNF-α, IL-1β, and IFN-γ, as well as proteinuria, and can improve fetal outcomes [81]. Another study showed that silibinin (70 mg/kg by injection) can reduce inflammation in human fetal membranes and myometrium by decreasing the expression of IL-6, IL-8, COX-2, and PGE_2_ and PGF_2α_. [76]. Moreover, silibinin showed hepatoprotective effects in different periods of pregnancy through improving the hepatic cellular architecture and the muscular wall if hepatic arteries are impaired by L-NAME [81]. However, it should be noted that Ali et al. [88] demonstrated in vitro that silibinin inhibited the cardiac differentiation of embryonic stem cells; thus, according to the authors, the use of this chemical compound during pregnancy should be avoided [88].

One study [86] tested three doses of vitexin (30, 45, 60 mg/kg) in pregnant rats with experimental preeclampsia induced by NG-nitro-L-arginine-methyl-ester; a protective effect against preeclampsia damage was observed, which decreases the high systolic blood pressure dose- and time-dependently. Vitexin also reduced placental TFPI-2, HIF‑1α, and VEGF during experimental preeclampsia. This flavonoid can reduce oxidative stress in the blood and placenta. According to Babaei et al. [89], vitexin as an anti-oxidant molecule exerts protective activity against various reactive oxygen species and lipid peroxidation, which are involved in pathophysiological pathways during preeclampsia. Furthermore, vitexin decreased the expression levels of sFlt and PlGF, which play key roles in the pathogenesis of preeclampsia. Also, vitexin at a dose of 60 mg/kg was more effective in low pups/placenta ratio [86]. All results have been summarized in Table 1.

## 4. Polyphenolic Compounds in Clinical Trials and Case-Control Studies

Recently, a few reviews of results from clinical trials concerning the effectiveness and safety of natural chemical compounds in pregnant women with hypertension/preeclampsia have been carried out [65,90]. However, clinical evidence on the effects of polyphenolic compounds in preeclampsia is still lacking.

Fadinie et al. [91] drew attention to the anti-inflammatory properties of curcumin, which can inhibit the activation of NF-κβ and may influence the activity and concentrations of COX-2 and IL-10, involved in the pathogenesis of preeclampsia. In order to verify this assumption, Fadinie et al. [91] carried out a double-blinded, randomized clinical trial with pregnant women with preeclampsia (*n* = 47). Patients received medicinal products containing 100 mg of curcumin once daily (*n* = 23). However, this study did not show statistically significant differences in levels of analyzed markers in serum of patients after treatment with curcumin.

Other authors [92] studied epigallocatechin gallate (100 mg in five doses), which was administered to pregnant women with severe preeclampsia (*n* = 304 patients) in a double-blind, randomized, placebo-controlled clinical trial. The results were also compared with a group that took nifedipine (10 mg in five doses) with epigallocatechin gallate (100 mg) (*n* = 148 patients) or each of these medicines alone. The study showed that the combination of two drugs in therapy was therapeutically efficacious without differences between groups and it was safe. Moreover, epigallocatechin gallate reduced the side effects of nifedipine, that is, vomiting and hypotension.

The results of a randomized clinical trial with a similar study design as above [93] showed that resveratrol (50 mg, up to five dosages) was an effective adjuvant of nifedipine (10 mg, up to five dosages) in the treatment of women with diagnosed preeclampsia (*n* = 174). Adverse effects did not differ between groups. However, it should be noted that administration of resveratrol with nifedipine reduced vomiting and hypotension, too.

Another study evaluated the influence of silibinin on the expression of various genes involved in inflammatory processes and levels of cytokines in monocytes obtained from preeclamptic women [94]. It was observed that administration of silibinin effectively increased expression of *IL-10*, reduced activation of the inflammasome (*NLRP1*, *NLRP3*, *Caspase-1*) and gene expression of the NF-κB-pathway decreased NF-κB levels and diminished production of IL-1β, IL-18, and TNF-α. Previously, other studies showed similar effects in peripheral blood mononuclear cells obtained from women with preeclampsia after intervention with the use of silibinin including the inhibition of activation of the NF-κB-pathway and production of pro-inflammatory cytokines [80,95]. All results have been summarized in Table 2.

## 5. Plant Extracts in Animal Models of Preeclampsia

It is well known that plant extracts as a complex of various chemical compounds can be a promising source of pharmacologically active substances for future use in therapies, especially in preeclampsia. These cocktails of active ingredients may show properties through synergy in their action [48,96,97]. Recently, five plant extracts, *Euterpe oleracea*, *Moringa oleifera*, *Thymus schimperi*, *Uncaria rhynchophylla*, *Vitis vinifera*, have been investigated in animal models of preeclampsia.

*Euterpe oleracea* Mart. from Arecaceae grows in many countries of South America (i.e., Brazil, Ecuador, and Venezuela) and is a source of nutritious fruits (acai berry), leaves, and seeds [98,99,100]. Phytochemical studies showed that extract from seeds of *E. oleracea* contain polyphenolic compounds, for example, proanthocyanidins, epicatechin, catechin [101]. Extracts from fruits include quercetin, catechin, procyanidin oligomers, and phenolic acids (ferulic acid, vanillic acid, syringic acid) [100]. Various preparations from different parts of this plant exert a wide range of pharmacological activities, mainly tissue-protective, antioxidative, anti-inflammatory, anti-hyperlipidemic, and anti-hypertensive effects [99,100]. Recently, a study performed by Da Silva et al. [99] assessed the cardiovascular effects of an extract containing polymeric proanthocyanidins from seeds of *E. oleracea* in a preeclampsia animal model after application of L-NAME (60 mg/kg/day). The extract was administered to rats at a dose of 200 mg/kg/day in drinking water between days 13 and 20 of gestation and there was observed decreasing blood pressure which was elevated by L-NAME in the second half of pregnancy. Other results of supplementation of the extract were anti-oxidant effects, which included a decreasing level of lipid peroxidation products, diminishing maternal microalbuminuria, and increasing total placental mass and fetal weight. Additionally, the extract significantly elevated effects of the vasodilator responses after application of agonists (acetylcholine, angiotensin II, and bradykinin) in the isolated mesenteric arterial bed, although it did not change nitrite content (NO). Effects of extract of *E. oleracea* may open a new window for discovering multi-targeting drugs in the prevention and treatment of preeclampsia.

The next plant is *Moringa oleifera* Lam., the most cultivated species, which is naturally occurring in various regions of India [102]. All parts of this plant contain many valuable secondary metabolites such as flavonoids (e.g., quercetin, kaempferol, isorhamnetin) [103], phenolic acids (e.g., chlorogenic acids) [104], glucosinolates (e.g., glucomoringin) [105], and tannins [106]. These compounds show a wide range the pharmacological activities, mainly anti-oxidant, anti-inflammatory, anti-cancer, and anti-diabetic properties [102]. Recently, Batmomolin et al. [107] carried out a study concerning the efficacy of ethanolic extract from the leaves of *M. oleifera* in a rat model of preeclampsia induced by L-NAME (50 mg/kg b.w.), which was applied on days 5–18 of gestation. The plant extract was applied to rats in various doses (50, 100, and 200 mg/kg b.w.) and the results were compared with acetylsalicylic acid (1.35 mg/200 g b.w.). Batmomolin et al. [107] reported that two doses of extract of *M. oleifera* (50 and 100 mg/kg) showed stronger activity than low-dose acetylsalicylic acid in decreasing the concentration of IL-17 in the serum of rats. Moreover, all the doses of extract (50, 100, 200 mg/kg) exerted a similar preventive effect as the low-dose acetylsalicylic acid against elevation of systolic and diastolic blood pressures. The extracts showed a tendency to reduce the level of soluble vascular endothelial growth factor receptor 1 (sFlt-1).

Recent studies have shown promising activity of extract of *Thymus schimperi* Ronniger in animal models of preeclampsia [108]. This species belongs to the Lamiaceae family, is a representative plant of Ethiopia [109], and contains mainly essential oil (i.e., β-myrcene, α-terpinene, limonene, α-humulene, p-cymene) [110]. Various extracts of *T. schimperi* have shown anti-oxidant potential [111], anti-hyperglycemic properties [109], and anti-hypertensive activity [112]. Mergiaw et al. [108] firstly studied the effect of aqueous crude extract from leaves of *T. schimperi* at doses of 250, 500, and 1000 mg/kg/day administered to rats with preeclampsia induced by L-NAME (50 mg/kg, p.o.). The results showed that values of red blood cells after extracts were given in a dose-dependent manner were similar to the control group, which received nifedipine at a dose of 20 mg/kg/day. Moreover, all doses of extracts reduced the levels of hemoglobin and hematocrit and increased platelets and total leukocyte count. In the case of these hematologic parameters, the greatest beneficial effect was observed after 1000 mg/kg of extract. For this reason, the extract of *T. schimperi* may be considered in relieving the effects of preeclampsia.

Another valuable plant is *Uncaria rhynchophylla* (Miq.) Miq. ex Havil. (*Rubiaceae*), naturally occurring in tropical regions, which contains indole alkaloids (rhynchophylline, isorhynchophylline), phenolic compounds (hyperin, epicatechin, caffeic acid, procyanidin B2), and pentacyclic triterpene esters [113,114]. Extracts showed anti-oxidant, antimutagenic and cytoprotective activities [113], and anti-inflammatory actions [114]. According to traditional Chinese medicine, this plant is used in the treatment of hypertension and headache [114]. An anti-hypertensive effect of *U. rhynchophylla* was also observed in spontaneously hypertensive rats [115]. Previously, it was also shown that *U. rhynchophylla* inhibited the synthesis of nitric oxide and interleukin IL-1b induced by LPS in macrophage culture [116]. Inflammatory markers such as cytokines (IL-1b, IL-6, TNF-α, INF-γ) are involved in the pathogenesis of preeclampsia [117]. Wu et al. [118] tested various doses of extract from *U. rhynchophylla* (35, 70, and 140 mg/kg b.w./day) in a rat model of preeclampsia induced by a lipopolysaccharide (LPS; 1 mg/kg b.w./day). The extract contained isorhynchophylline, yohimbine, 3α-dihydrocadambine, raubasine, hirsuteine, and hirsutine, and was administered to rats between 14 and 19 days of gestation. The results showed that the extract at the highest dose reduced systolic blood pressure levels between 14 and 18 days of gestation. Values of systolic blood pressures were lower after all doses of the extract in comparison with the group which was given only LPS. Further analysis demonstrated that the level of urinary albumin was decreased after application of the extract in a dose-dependent manner. Levels of serum and placental cytokines (IL-6, IL-1b, TNF-a, IFN-g) were significantly diminished after the highest dose of extract, similarly as mRNA expression of pro-inflammatory cytokines and the level of NF-κB p65 in the placenta. Moreover, the number of live fetuses was higher after the use of the extract at a dose of 140 mg/kg. The results suggest that alkaloid extract of *U. rhynchophylla* may be a novel therapeutic for preeclampsia, although further studies are needed [118].

Another interesting plant in this field is *Vitis vinifera* L., extracts from it, containing potent polyphenol anti-oxidants (e.g., proanthocyanidins, stilbenoids), showed cardioprotective, vasoprotective, and anti-inflammatory effects [119,120,121]. For this reason, extracts from skins of *V. vinifera* may be a promising source of natural substances in the prevention and treatment of preeclampsia. Previously, de Moura et al. [122] studied the activity of extract from the skins of *V. vinifera* fruits in experimental preeclampsia induced by L-NAME (60 mg/kg/day, p.o.). This extract was applied to pregnant rats at a dose of 200 mg/kg/day from 13 to 20 days of pregnancy. This intervention showed that the extract prevented increasing arterial pressure and insulin resistance in comparison with the group with experimental preeclampsia. The vasodilatory effect of the extract is taken into account in its mechanism of action. Recently, Da Costa et al. [123] observed that *Vitis vinifera* L. grape skin extract containing peonidin3-*O*-glucoside, petunidin-3-*O*-glucoside, malvidin-3-*O*-glucoside, and malvidin-3-(6-*O*-trans-p-coumaryl)-5-O-diglicoside showed a preventive effect against increasing systolic blood pressure. The extract was tested at a dose of 200 mg/kg/day in drinking water for 12 weeks in spontaneously hypertensive rats (SHR). According to the authors, the beneficial effect of the extract is dependent on increasing anti-oxidant activity of superoxide dismutase observed in the plasma and kidneys of animals after treatment with the extract. Furthermore, Moura et al. [124] found that extract from skin of fruits of *Vitis labrusca* (red grapes) at a dose of 100 mg/kg exerted not only vasodilatory and anti-oxidant, but also anti-hypertensive effects. The reduction of systolic and diastolic pressures was observed in rats with hypertension induced by L-NAME (50 mg/kg/day, p.o.). A similar effect was observed in a group with hypertension induced by desoxycorticosterone acetate (DOCA). Thus, it seems that the mechanism of action of the extract does not involve the regulation of the renin-angiotensin system. All results have been summarized in Table 1.

## 6. Plant Extracts in Clinical Trials and Ex Vivo Studies of Preeclampsia

More and more plant extracts are considered as forward-looking ways to improve outcomes of preeclampsia therapy. Promising plants studied in this regard are *Brassica oleracea* L. var. *Italica* Plenck and *Silybum marianum* (L.) Gaertner. These studies are very much needed as there are currently a limited number of therapeutic interventions for mild and severe preeclampsia.

*Brassica oleracea* L. var. *Italica* Plenck (Brassicaceae) contains glucosinolates (e.g., sinigrin), isothiocyanates (e.g., sulforaphene), phenolic compounds (e.g., caffeic and chlorogenic acids, apigenin, luteolin, kaempferol, quercetin), and carotenoids [125,126]. Broccoli sprouts have a wide range of beneficial and health-promoting activities, mainly anti-oxidant, antimicrobial, anti-inflammatory properties [125,127]. Besides the great economic importance of *Brassica oleracea* L. var. *Italica*, in recent times, extract from broccoli sprouts is considered as a source of new medicines and it is proposed to be investigated in a clinical trial (phase III) concerning a promising intervention in women with early-onset preeclampsia [128]. In a double-blind, placebo-controlled randomized study, the commercial extract will be given to patients as an adjuvant therapy at a dose of three capsules twice daily (total of 24 mg of activated sulforaphane) or placebo. The authors will study the influence of the sulforaphane compounds on biomarkers of the function of the placenta and endothelium, and maternal and fetal outcomes. The clinical trial is in progress (No. ACTRN12618000216213).

*Silybum marianum* (L.) Gaertner from the Asteraceae family is a very important medicinal plant used since ancient times, whose seeds and fruits are mainly used in the prevention and treatment of liver diseases and gallbladder disorders due to the content of silymarin consisting of several flavonolignans, mainly silibinin (or silybin) [74], which is the most investigated active component [80,94,95]. To date, only one clinical trial has been conducted on the treatment of preeclampsia using extracts of *S. marianum* [129,130]. Other studies concerned the biological material collected from women with preeclampsia [80,94,95]. In these interventions, anti-oxidant, cytoprotective, and immunomodulatory activities of chemical compounds of *S. marianum* were considered. In a randomized, double-blinded clinical trial by Baghbahadorani et al. [130] (No. IRCT201509042388/N1), 60 women with severe preeclampsia were studied. In order to evaluate the effect of silymarin (an extract of *S. marianum*) on platelet abnormalities, it was given to patients at a dose of 70 mg twice, 3 h and 24 h after birth. However, the results did not show a significant difference between intervention and control (placebo) groups in the number of platelets or in other factors. Further analysis [129] of the results of this clinical trial showed that silymarin had an influence on liver enzymes (AST, ALT, ALP) and it showed a decreasing trend of ALT level. There is a need for further studies using other durations of intervention and dosage of silymarin extract. All results have been summarized in Table 2.
pharmaceuticals-14-00269-t001_Table 1Table 1Summary of studies on pharmacological effects of plant-derived polyphenols and plant extracts in animal models of preeclampsia.Polyphenols/Plant ExtractsModels and DosesEffectsRef.Baicalin- preeclampsia induced by L-NAME (50 mg/kg b.w./day) in female Sprague–Dawley rats (*n* = 60),- doses of baicalin: 50, 100, and 150 mg/kg/day, i.p. for 20 days.- alleviating the blood pressure in a dose-dependent manner,- decreasing the apoptosis of cells of kidney and livers,- increasing the expressions of antiapoptotic protein (XIAP and Bcl-2),- reducing the urinary protein level.[13]Curcumin- preeclampsia induced by LPS (injection of 0.5 μg/kg) in female Sprague–Dawley rats (*n* = 14),- dose of curcumin: 0.36 mg/kg (injection after LPS administration).- decreasing the blood pressure and urinary protein level,- improving the deficient trophoblast invasion and spiral artery remodeling,- decreasing the TLR4, p65, JunB protein expressions in placenta,- decreasing the IL-6 (by 23.42%) and MCP-1 mRNA expressions (52.67%) in serum and placenta.[18]- preeclampsia induced by LPS (injection of 10 μg/kg) in mice (*n* = 20),- dose of curcumin: 0.5 mg/kg (i.g.) for 17 gestational days.- decreasing the systolic blood pressure and proteinuria, - increasing the number of live pups, fetal weight, placental weight, - decreasing the fetal resorption rate, - suppressing the placental expressions of TNF-α, IL-1β, IL-6, - upregulation of the phosphorylated Akt level in placenta after curcumin.[19]Punicalagin- preeclampsia induced by L-NAME (50 mg/kg b.w./day) in female Sprague–Dawley rats (*n* = 40),- doses of punicalagin: 25, 50, or 100 mg/kg orally on days 14–21 of pregnancy.- decreasing systolic and diastolic blood pressure and also mean arterial pressure, - increasing the expression of vascular endothelial growth factor, - downregulating vascular endothelial growth factor receptor-1/fms-like tyrosine kinase-1.[13]Quercetin- preeclampsia induced by LPS (1.0 infusion of 1.0 μg/kg) in female Sprague–Dawley rats (*n* = 24),- dose of quercetin: 2 mg/kg b.w.- significant reduction of the systolic blood pressure by 15%, - decreasing the elevated changes of tyrosine kinase-1 (sFlt-1)/placental growth factor (PlGF) ratio, - suppressing the production of cytokine production in the placenta (TNF α, IL-6, and MCP-1), - reduction of lipid peroxidation by reducing the MDA level,- no difference of fetus size in group with and without supplementation of quercetin,- increasing the weight of placentas reduced by LPS.[45]- preeclampsia induced by L-NAME (0.5 mg/mL in drinking water) in female Sprague–Dawley rats (*n* = 40),- dose of quercetin: 2.0 mg/kg b.w. by intraperitoneal infusion and acetylsalicylic acid (1.5 mg/kg b.w.) in rodent dough.- reducing the systolic blood pressure and proteinuria (quercetin enhanced the effect of acetylsalicylic acid), - decreasing expressions of mRNA VEGF and mRNA sFlt-1,- reduction of lipid peroxidation by reducing the MDA level, - reducing of IL-6 and TNF-α levels, All effects were strongest in the group supplemented by quercetin with acetylsalicylic acid.[46]- preeclampsia induced by L-NAME (50 mg/day; i.p.) in female Sprague–Dawley rats (*n* = 30),- dose of quercetin: 10 mg/kg b.w. i.p.- no effect on decreasing the high blood pressure,- normalized in proteinuria.[47]Resveratrol- preeclampsia induced by L-NAME (125 mg/kg b.w.; injection) in female albino Wistar mice,- dose of resweratrol: 20 mg/kg/day i.g.- reducing the systolic blood pressure and urine protein level compared with the L-NAME group, - decreasing the expression of mRNA sFlt-1 compared with the L-NAME group, - increasing the expression of mRNA VEGF, AngI, and AngII compared with the L-NAME group, - activation of epithelial-mesenchymal transition.[60]- preeclampsia induced by L-NAME (125 mg/kg b.w.; injection) in female Wistar albino rats- dose of resveratrol: 20 mg/kg per day, i.g. during the entire pregnancy.- reducing the systolic blood pressure and levels of protein/creatinine,- anti-apoptotic effects in trophoblasts of placentas.[62]- preeclampsia induced by desoxycorticosterone acetate (12.5 mg by injection) in female Wistar albino rats- dose of resweratrol: 40 mg/kg per day, i.g.- no effect on decreasing the high blood pressure, placental and renal blood flows,- no effect on placental pathology parameters.[63]Salvianolic acid A- preeclampsia induced by phosphatidyleserine/phosphatidylcholine (100 μL in suspension; i.p.) in mice- dose of salvianolic acid A: 10 μg/g and 30 μg/g, i.p.- reducing the thrombin time similarly (as heparin), - increasing the plasma antithrombin III activity (as acetylsalicylic acid), - high-dose of salvianolic acid A more effective in decreasing blood pressure to normal level, - high-dose of salvianolic acid A more effective in decreasing proteinuria to normal level.[73]- preeclampsia induced by phosphatidyleserine/phosphatidylcholine (100 μL in suspension; i.p.) in mice- dose of salvianolic acid A: 10 μg/g and 30 μg/g, i.p.- high-dose of salvianolic acid A more effective in decreasing blood pressure to normal level, - high-dose of salvianolic acid A more effective in decreasing proteinuria to normal level,- lowering the expression of thrombomodulin in placenta.[72]Silibinin- preeclampsia induced by L-NAME (70–80 mg/kg/day in drinking water) in female Wistar rats- dose of silibinin: 100 mg/kg/day by 10 days (by gavage).- reducing the systolic blood pressure, - reducing the level of pro-inflammatory factors: TNF-α, IL-1β, IFN-γ, - reducing the proteinuria,- normalized the platelet count, - improving the fetal outcomes.[81]- preeclampsia induced by LPS (50 mg/mouse; i.p.) in C57BL/6 mice,- dose of silibinin: 70 mg/kg by injection.- decreasing the expression of IL-6, IL-8, MMP-9, - decreasing the expression of IL-6, IL-8, COX-2, PGE2, PGF2a.[76]Vitexin- preeclampsia induced by L-NAME (0.5 mg/mL in drinking water) female Sprague–Dawley rats,- dose of vitexin: 30, 45, 60 mg/kg for 10 days.- reducing the systolic blood pressure, - diminishing TFPI-2, HIF 1α, and VEGF in placenta,- alleviating the oxidative stress in blood and placentas,- high dosage (60 mg/kg) decreased sFlt-1, increased PlGF, - dose of 60 mg/kg more effective in low pups/placenta ratio.[86]*Euterpe oleracea* aqueous crude extract from seeds (polymeric proanthocyanidins, catechin, epicatechin)- preeclampsia induced by L-NAME (60 mg/kg/day in drinking water) in female Wistar rats- dose of extract: 200 mg/kg in drinking water during 8 days.- reducing the blood pressure in the second half of pregnancy, - decreasing of lipid peroxidation, - diminishing maternal microalbuminuria, - increasing in total placental mass, and fetal weight,- no effect in activities of superoxide dismutase, catalase, glutathione peroxidase,- lowering the nitrite content (NO).[99]*Moringa oleifera *- ethanolic extract from leaves (e.g., flavonoids—quercetin)- preclampsia induced by L-NAME (50 mg/kg/day) - doses of extract: 50,100, and 200 mg/kg b.w, during 13 days of gestation.- reducing the systolic and diastolic blood pressure (all doses) - similar preventive effect as the low-dose acetylsalicylic acid (1.35 mg/200 g b.w.),- decreasing the concentration of IL-17 (after extract at doses 50 and 100 mg/kg).[107]*Thymus schimperion*- aqueous crude extract from leaves- preeclampsia induced by L-NAME (50 mg/kg/day) - doses of extract: 250, 500, and 1000 mg/kg/day during nine days of gestation.- decreasing the levels of hemoglobin and hematocrit (all doses),- increasing count of platelets and total leukocyte,- the highest effect after 1000 mg/kg of extract.[108]*Uncaria rhynchophylla*- hydroethanolic extract (oxindole alkaloids: isorhynchophylline, yohimbine, 3α-dihydrocadambine, raubasine, hirsuteine, hirsutine)- preeclampsia induced by LPS (1.0 mg/kg b.w./day; injection) in female Sprague–Dawley rats,- doses of extract: 35, 70, and 140 mg/kg b.w./day during six days.- reducing the systolic blood pressure between 14 and 18 days of gestation (after extract at a dose of 140 mg/kg), - decreasing the level of urinary (after extract in a dose-dependent manner),- diminishing the levels of serum and placental cytokines: IL-6, IL-1b, TNF-a, IFN-g (after extract at a dose of 140 mg/kg),- diminishing the mRNA expression of pro-inflammatory cytokines in placenta, - diminishing the level of NF-jB p65 in the placenta,- higher the live fetuses (after extract at a dose of 140 mg/kg).[118]*Vitis labrusca*- hydroethanolic extract from skin of fruits (polyphenols concentration 55.5 mg g­ 1)- hypertension induced by L-NAME (60 mg/kg/day, in drinking water, for 28 days) in male Wistar rats,- dose of extract: 100 mg/kg/day during 28 days of pregnancy.- increasing the heart rate,- decreasing the systolic, mean, and diastolic arterial pressure after four weeks after administration of extract,- decreasing the lipid peroxidation in liver.[124]*Vitis labrusca*- hydroethanolic extract from skin of fruits (polyphenols concentration 55.5 mg g­ 1)- hypertension induced by deoxycorticosterone acetate (12.5 mg kg ^−1^ per week, drinking solution for 30 days); male Wistar rats,- dose of extract: 100 mg/kg/day during 13 days of pregnancy.- increasing the heart rate,- decreasing the systolic, mean, and diastolic arterial pressure - decreasing the lipid peroxidation in liver.[124]*Vitis vinifera* grape skin extract- hypertension induced by L-NAME (60 mg/kg/day, in drinking water, for seven days) in male Wistar rats,- dose of extract: 200 mg/kg/day for seven days.- preventing the increasing the arterial pressure and insulin resistance.[122]*Vitis vinifera* grape skin extract- spontaneously hypertensive rats,- dose of extract ACH09: 200 mg/kg/day in drinking water for 12 weeks.- reducing the systolic blood pressure,- decreasing the elevated concentrations of cholesterol and triglyceride, - diminishing the formation of products of peroxidation of lipid,- no effect in catalase activity.[123]
pharmaceuticals-14-00269-t002_Table 2Table 2Summary of studies on pharmacological effects of plant-derived polyphenols and plant extracts in clinical trials and ex vitro model.Polyphenols/Plant ExtractsStudy DesignEffectsRef.Curcumin- double-blind, randomized clinical trial,- pregnant women with preeclampsia (*n* = 47),- dose of curcumin: 100 mg once daily.- no significant differences in level of markers in serum of patients.[91]Epigallocatechin gallate- double-blind, randomized, placebo-controlled clinical trial,- pregnant women with severe pre-eclampsia (*n* = 304 patients),- dose of epigallocatechin gallate:(1) 100 mg with 10 mg nifedipine,(2) 100 mg without nifedipine.- more effective of the combination of two drugs in therapy,- reducing time needed to control blood pressure after combinational treatment,- lower number of treatment doses needed to control blood pressure after combinational treatment,- decreasing the side effects of nifedipine, i.e., vomiting and hypotension after epigallocatechin gallate.[92]Resveratrol- double-blind, randomized, placebo-controlled clinical trial,- pregnant women with severe pre-eclampsia (*n* = 349 patients),- dose of resveratrol: 50 mg (up to five dosages) with nifedipine: 10mg (up to five dosages).- more effective of the combination of two drugs in therapy,- reducing time needed to control blood pressure after resveratrol with nifedipine,- lower number of treatment doses needed to control blood pressure after combinational treatment,- decreasing the side effects of nifedipine, that is, vomiting and hypotension after resveratrol.[93]Silibinin- 20 women with diagnosed preeclampsia,- collected blood samples from women with preeclampsia,- monocyte preparations cultured in the presence of silibinin: 50 µM.- increasing the expression of IL-10,- reduced the activation of inflammasome (*NLRP1*, *NLRP3*, *Caspase-1*) and gene expression of NF-κB-pathway,- decreased the NF-κB levels,- decreasing the production of IL-1β, IL-18, and TNF-α.[94]- 30 women with diagnosed preeclampsia,- collected blood samples from women with preeclampsia,- peripheral blood mononuclear cells cultured in the presence of silibinin: 5 µM and 50 µM.- decreasing the production of TNF-α after silibinin at a concentration of 50 µM,- inhibiting the spontaneous releasing the reactive oxygen species (superoxide anion and hydrogen peroxide anion).[95]- 30 women with diagnosed preeclampsia,- collected blood samples from women with preeclampsia,- peripheral blood mononuclear cells cultured in the presence of silibinin: 5 µM and 50 µM and LPS-stimulated cells.- decreasing the NF-κB activity after silibinin at a concentration of 50 µM,- diminishing the production of TNF-α and IL-1β after silibinin at a concentration of 5 µM and 50 µM.[80]*Broccoli sprout extract*- double-blind, placebo-controlled randomized study (phase III)—in progress.- not yet available.[128]*Silybum marianum*- double-blind, randomized, placebo-controlled clinical trial,- women with severe preeclampsia (*n* = 60),- silymarin (the extract of *Silybum marianum*) at a dose of 70 mg twice, three hours after birth and 24h later.- influencing the liver enzymes (AST, ALT, ALP),- decreasing the level of ALT (trend).[129]

## 7. Conclusions

Our review highlights that most of the evaluations of the mechanisms of action of selected plant phenolic compounds and plant extracts in preeclampsia animal models have been carried out only in recent years. This indicates a new and very important trend for pharmacological studies and the future applicability of their results during the intensive search for new therapeutic solutions for this very dangerous disease for the mother and fetus. However, there are still too few of these studies and clinical trials to allow definitive conclusions to be drawn. Nevertheless, compounds such as baicalin, curcumin, epigallocatechin gallate, punicalagin, quercetin, resveratrol, salvianolic acid A (danshensu), silibinin, and vitexin are potential and promising candidates for further research. All of them have a beneficial anti-inflammatory effect at the molecular and cellular level, which is of great importance in the pathogenesis of preeclampsia. It should also be emphasized that the study of herbal extracts from *Brassica oleracea*, *Euterpe oleracea*, *Moringa oleifera*, *Punica granatum*, *Silybum marianum*, *Thymus schimperi*, *Uncaria rhynchophylla*, and *Vitis vinifera* is a future direction for rational phytotherapy as a complementary therapy during the treatment of preeclampsia. However, systematic studies in this area combining in silico, in vitro, and in vivo models are needed. Moreover, clinical studies on the mechanisms of action, as well as the effectiveness and safety of using plant phenolic compounds and herbal extracts during pregnancy are necessary to perform.

## Figures and Tables

**Figure 1 pharmaceuticals-14-00269-f001:**
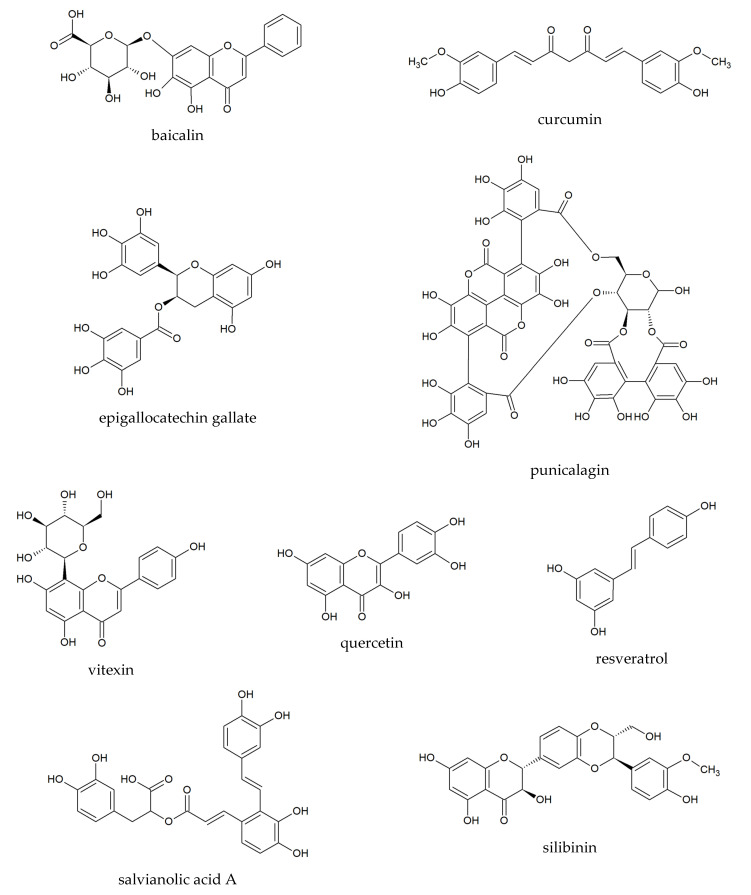
Chemical structures of polyphenolics.

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
