# Peer review of "Plant Phenolics and Extracts in Animal Models of Preeclampsia and Clinical Trials—Review of Perspectives for Novel Therapies"

_pharmaceuticals, 2021, doi:10.3390/ph14030269_

Round 1

Reviewer 1 Report

In my view, the authors have made a good work in revising the literature existing to date on the field. This review seems enough clear and well organized. Just few concerns about unclear parts and English revision, which I grouped together in the attached file.

Reviewer 2 Report

Type of the Paper (Review)

Title: Plant phenolics and extracts in animal models of pre-eclampsia and clinical trials – review of perspectives for novel therapies

Minor error:

       1-Comments: Page 2, Line 73: Reference[7,8] instead of [7, 8], no space between the number, please check the similar mistake throughout the manuscript

      2-Comments: Need to uniform (write the same way): anti-hyperglycemic and antihyperglycemic; antihypertensive and anti-hypertensive; anti-oxidant and antioxidant; anti-inflammatory and antiinflammatory

           2a) Line 415: antihyperglycemic; Line 118: anti-hyperglycemic

            2b) Line 30, 50, 84, 78, 160, 381: anti-hypertensive; Line 119, 142, 416, 193, 432, 471, 615, 620, 844, 876: antihypertensive

           2c) Line 180: anti-oxidant; Line 50 77, 109, 118,471, 489, 506, 568, 573, 576 609, 634, 666, 785, 842, 866, 876: antioxidant

Round 2

Reviewer 1 Report

The authors answered my concerns correctly, so I now believe the manuscript is worth publishing